# Zebrafish: A New Promise to Study the Impact of Metabolic Disorders on the Brain

**DOI:** 10.3390/ijms23105372

**Published:** 2022-05-11

**Authors:** Batoul Ghaddar, Nicolas Diotel

**Affiliations:** Diabète Athérothrombose Thérapies Réunion Océan Indien (DéTROI), INSERM, UMR 1188, Université de La Réunion, 97400 Saint-Denis, France; batoul.ghaddar@univ-reunion.fr

**Keywords:** brain plasticity, diabetes, metabolic disorders, neural stem cell, obesity, zebrafish

## Abstract

Zebrafish has become a popular model to study many physiological and pathophysiological processes in humans. In recent years, it has rapidly emerged in the study of metabolic disorders, namely, obesity and diabetes, as the regulatory mechanisms and metabolic pathways of glucose and lipid homeostasis are highly conserved between fish and mammals. Zebrafish is also widely used in the field of neurosciences to study brain plasticity and regenerative mechanisms due to the high maintenance and activity of neural stem cells during adulthood. Recently, a large body of evidence has established that metabolic disorders can alter brain homeostasis, leading to neuro-inflammation and oxidative stress and causing decreased neurogenesis. To date, these pathological metabolic conditions are also risk factors for the development of cognitive dysfunctions and neurodegenerative diseases. In this review, we first aim to describe the main metabolic models established in zebrafish to demonstrate their similarities with their respective mammalian/human counterparts. Then, in the second part, we report the impact of metabolic disorders (obesity and diabetes) on brain homeostasis with a particular focus on the blood–brain barrier, neuro-inflammation, oxidative stress, cognitive functions and brain plasticity. Finally, we propose interesting signaling pathways and regulatory mechanisms to be explored in order to better understand how metabolic disorders can negatively impact neural stem cell activity.

## 1. Introduction

Metabolic syndrome is a combination of at least three of the following metabolic disorders: abdominal obesity, hypertriglyceridemia, low serum high-density lipoprotein (HDL) levels, hyperglycemia (associated with insulin resistance) and hypertension [1]. These biochemical and physiological dysfunctions significantly increase the risk of chronic kidney disease, hepatic steatosis and cardiovascular diseases, including myocardial infarction and stroke, among others [2,3,4]. Recently, metabolic syndrome has also been suggested to be associated with cognitive and behavioral impairments [5].

Diabetes and obesity are two metabolic diseases related to metabolic syndrome. Both diseases are closely associated: approximately 84% of people with type 2 diabetes are also obese and/or overweight [6]. In addition, diabetes and obesity cause many deleterious effects on the body. For example, they strongly affect the renal glomerular filtration rate and increase the risk of developing chronic kidney disease [7]. They are also risk factors for the development of cardiovascular diseases affecting blood pressure and endothelial cell and cardiomyocyte functions [8,9]. In mice, obesity promotes adipose tissue inflammation, leading to liver inflammation and promoting glucose intolerance and insulin resistance [10]. Similarly, insulin resistance was correlated with an increased risk of hepatic steatosis in a Korean cohort, even before the onset of a diabetic state [11]. Overall, diabetes and obesity impair the cardiovascular, renal, visual, intestinal and metabolic systems.

Obesity and diabetes have been more recently documented for their deleterious consequences on the central nervous system, especially on cognitive processes [12,13]. Both diseases disrupt the blood-brain barrier (BBB) [14,15] and promote neuro-inflammation and cerebral oxidative stress [16]. Obesity and diabetes have also been suggested to be involved in neuronal degeneration, being risk factors for the development of neurodegenerative diseases, including Alzheimer’s disease [17]. This is likely related to increased oxidative stress through mitochondrial dysfunction and neuro-inflammation [18,19,20]. Strikingly, recent data have demonstrated that diabetes and obesity have a deleterious impact on brain plasticity and, among others, on neurogenesis.

Neurogenesis is an evolutionarily conserved process that involves the division of neural stem cells, the genesis of other committed progenitors and, finally, the birth of new neurons that can migrate and differentiate within the nervous tissue [21,22,23,24]. This interesting and intriguing process occurs primarily during development but has also been documented in the adult brain of all species studied to date, including humans [25,26]. In mammals, adult neurogenesis occurs mainly in two regions: (1) the subventricular zone (SVZ) of the lateral ventricle and (2) the subgranular zone (SGZ) of the dentate gyrus of the hippocampus [27,28]. Neurogenesis is tightly regulated by many different factors, including hormones and oxygen supply, as well as trophic, immune and epigenetic factors [29,30]. The microenvironment of the neurogenic niche can consequently disrupt neural stem cell activity, namely, under chronic and/or acute inflammation, as well as under pro-oxidant conditions [31,32,33,34,35,36].

To better understand the mechanisms underlying the deleterious effects of metabolic disorders in the brain of obese and/or diabetic individuals, several animal models have been used, such as non-human primate models (i.e., chimpanzee), large animal models (i.e., dog and pig), rodent models (i.e., rat and mouse) and non-mammalian models, such as nematode (*C. elegans*) and zebrafish (*Danio rerio*) [37]. Of course, working with large mammalian models has many drawbacks, including ethical concerns and difficult manipulations, as well as high costs. For these reasons, the main models used to study neurogenesis remain rodents. However, in recent years, zebrafish have emerged as an interesting model to study the impact of metabolic disorder on the brain [38,39,40,41,42,43].

Zebrafish is indeed an attractive organism to study metabolic disorders. This small teleost fish has metabolic organs conserved with humans, including the liver, adipose tissue, pancreas and kidney [44]. As in mammals, many methods are available to measure insulin, blood glucose, lipid and cholesterol levels in this small and easily manipulated model. Finally, over 70% of the human genome has orthologs in zebrafish, and many physiological processes are conserved between fish and mammals, including humans [45]. Thus, several groups have developed models of metabolic disorders to successfully mimic the human pathologies of diabetes and obesity. These include diet-induced obesity (DIO), high-fat diets (HFDs) and hyperglycemic/diabetic models, as well as genetic models of metabolic disturbances. Interestingly, the pathological disorders induced by hyperglycemia and/or obesity are parallel to those in humans [46,47,48]. In addition, zebrafish have almost unique characteristics when considering the central nervous system. The adult zebrafish brain retains a broad distribution of active neurogenic niches throughout the encephalon [22,49,50,51]. This is in striking contrast to mammals, in which neurogenic niches are restricted to two main regions, the SVZ and SGZ [21,27,28]. Furthermore, unlike mammals, the brain of adult zebrafish is able to regenerate efficiently after large lesions, without generating persistent glial scarring and without striking residual disabilities [21,52,53,54,55].

Today, more and more efforts are being made to better understand how metabolic disorders can alter brain regeneration in order to find new therapeutic approaches to combat their impacts on the CNS under constitutive and regenerative conditions. In this general context, zebrafish is a promising model that offers new hope to understand the disrupted mechanisms occurring during metabolic disorders. It also offers the possibility to discover new effective drugs to combat the deleterious effects induced by metabolic disruptions. In this review, we discuss the main zebrafish models developed to study the effects of obesity and diabetes on CNS functions. We then highlight, in a comparative approach, the deleterious effects of these models on brain homeostasis, focusing on the BBB and constitutive and regenerative neurogenesis, as well as on cognitive functions. Finally, we discuss potential mechanisms that could explain the deleterious effect of obesity and/or diabetes on the CNS.

## 2. Different Models of Metabolic Disorders in Adult Zebrafish

A growing number of studies has used zebrafish to investigate its relevance to hyperglycemia/diabetes as well as overweight/obesity. In this first section, we document the main models of metabolic disorders developed in zebrafish, focusing mainly on the adult stages.

### 2.1. Models of Acute and Chronic Hyperglycemia

**Acute hyperglycemia:** An interesting model aimed to develop acute hyperglycemia in fish by intraperitoneal injection of D-glucose (2.5 g/kg body weight) [40,56]. Such an injection resulted in a rapid and transient increase in blood glucose compared to zebrafish injected with vehicle. Thus, 1.5 h after D-glucose injection, blood glucose levels reached high values compared with control-injected fish (350–500 mg/dL glucose versus 100–150 mg/dL, respectively) [40,56]. In mice, a similar injection can be performed to study the impact of acute hyperglycemia in stroke models [57,58], or in cerebral blood flow and tissue oxygen saturation [59].

**Chronic hyperglycemia:** Other hyperglycemic models aimed to induce chronic and persistent hyperglycemia. In mammals, chronic hyperglycemia is mainly due to an alteration in insulin production, secretion and signaling. The zebrafish pancreas is quite similar to that of mammals with the presence of endocrine islets and, in particular, insulin-producing pancreatic β-cells [60]. Two main methods have been used to establish a larger and stable diabetic state: (1) the induction of hyperglycemia by the destruction of pancreatic β-cells (relative to type 1 diabetes) and (2) the induction of chronic hyperglycemia by dissolving D-glucose in fish water (relative to type 2 diabetes).

Type 1 diabetes can be induced in fish by pancreatectomy [61,62] or chemical-dependent ablation [63,64]. The former method is difficult to perform and not really used in the zebrafish community. In contrast, chemical ablation by intraperitoneal injection of drugs such as streptozotocin (STZ) and alloxan are widely developed [63,64]. These drugs, which have been widely used in mammals [65], lead to the death of pancreatic β-cells through the generation of oxidative stress. This results in impaired insulin production and high fasting blood glucose levels [66]. In zebrafish, several studies have demonstrated that injection of STZ and/or alloxan leads to hyperglycemia in larvae and adults [61,64,67,68,69]. For example, Olsen and colleagues showed that a serial injection of STZ in adult zebrafish leads to an increase in fasting blood glucose compared to control fish (~300 mg/dL vs. 60 mg/dL) from week 1 to 3 [67]. This treatment also induces higher levels of serum glycated protein (over 300%) and a ~80% reduction in insulin levels [67]. Interestingly, after 3 weeks, these hyperglycemic fish develop renal and retinal defects, as revealed by increased glomerular basement membrane thickness and decreased retinal layer thickness [67]. This is similar to the human situation, in which type 1 diabetic patients suffer from increased serum protein glycation levels [70,71], increased glomerular basement membrane thickness [72] and retinal complications [73]. Interestingly, hyperglycemic zebrafish also exhibit a reduced ability to regenerate their caudal fin after transection [67]. This interesting feature parallels the wound-healing defects observed in diabetic patients [74]. In other experiments using STZ, hyperglycemia has a deleterious impact on the cardiovascular system, leading to the misexpression of important cardiac proteins (P53, Ampk and Klf2a) and to the loss of cardiac myofibrils and their apoptosis, as well as to cardiac dysfunction [75]. In these hyperglycemic fish, stroke volume, cardiac output and ejection fraction (end-diastolic volume minus end-systolic volume) are lower than in control fish [75]. In addition, the expression of glucose transporters (GLUTs) is decreased in the heart of zebrafish, indicating a decreased ability of the zebrafish myocardium to utilize glucose [75].

Overall, these type 1 diabetes models exhibit many features of diabetic pathology in mammals (including humans), such as increased fasting blood glucose, increased plasma protein glycation, retinal and renal dysfunctions and cardiovascular complications, as well as impaired regenerative processes [67,76,77]. However, these diabetic models using STZ and alloxan have limitations, given the ability of fish to regenerate pancreatic β-cells over time [77,78,79].

Models of type 2 diabetes have been developed by supplementing fish water with D-glucose using different concentrations of D-glucose (55, 111 and 133 mM) [42,56,80]. However, young zebrafish (4–11 months) can acclimate better to glucose immersion than old fish (1–3 years) [81]. Although the different glucose concentrations lead to significant hyperglycemia between day 2 and day 3 of treatment, the 111 mM concentration remains the frequently used one in the literature [42,56]. Immersion of zebrafish in 111 mM D-glucose solution significantly increases blood glucose levels from nearly 3 mM (54 mg/dL) to 12 mM (216 mg/dL) in control and hyperglycemic fish, respectively [42]. Accordingly, our own experiments also demonstrated the significant increase in blood glucose levels after 14 days of treatment, from 60 mg/dL to 280 mg/mL [56,82]. Interestingly, this diabetic state is dependent on D-glucose dissolved in water, as a 7-day washout after 2 weeks of D-glucose treatment is sufficient to return to normal blood glucose levels [42]. This model also leads to the induction of hyperinsulinemia and impaired glucose metabolism, as well as higher glycation of ocular proteins, altered expression levels of insulin receptors in skeletal muscle and decreased blood glucose levels after treatment with antidiabetic drugs [42].

These general characteristics are commonly observed in diabetic mammalian models and human patients [83,84,85]. Type 2 diabetic rodents correspond mainly to (1) NOD mice (non-obese diabetic mice), (2) ob/ob and db/db mice with a mutation on leptin signaling components (leptin and its receptor, respectively) and (3) HFD or DIO models. In all these models, insulinemia, hyperglycemia and increased levels of serum protein glycation are observed. Overall, this work reflects the reliability of using hyperglycemic zebrafish to mimic human pathology.

**Genetic and transgenic models of hyperglycemia:** The use of genetic tools such as morpholinos, CrispR-Cas9 targeted gene ablation and transgenic and/or mutant lines has also allowed to generate hyperglycemic larvae and adult fish. For example, the overexpression of *foxn3*, a gene associated with fasting blood glucose regulation, leads to increased hepatic gluconeogenesis and fasting blood glucose in adulthood [86,87]. In adult zebrafish, knocking out the *pdx1* gene (a gene implicated in the development of type 2 diabetes) results in a reduced number of pancreatic β-cells, in decreased insulin levels and, consequently, hyperglycemia. It also delays body growth in fish [88]. Another transgenic model was achieved by overfeeding insulin-resistant skeletal muscle (zMIR) fish mutated on the IGF1 receptor [89]. Adult zMIR fish have normal glucose levels similar to control fish, but glucose levels increase significantly after the fish are overfed and describe the transitional state between insulin resistance and the development of type 2 diabetes [89]. Many other transgenic fish have been used, such as *deiodinase 2* KO and *aldh3a1* KO [90]. In addition to all these models, many diabetic and hyperglycemic protocols have been developed in the larva [90,91].

### 2.2. Models of Overweight and Obesity in Zebrafish Leading to Hyperglycemia

Obesity and overweight are characterized by hypertrophy and hyperplasia of adipocyte cells, resulting in increased body weight and body mass index (BMI). Zebrafish share with mammals the major metabolic organs regulating energy homeostasis (intestine, liver, pancreas, adipose tissue and muscle). Their respective functions are also evolutionarily conserved among taxa, including the regulation of feeding behavior, lipid storage and insulin secretion, among others [91]. Interestingly, the conserved metabolic pathways in adipocytogenesis and cholesterol metabolism between zebrafish and humans have made zebrafish an appropriate and alternative model in the field of metabolic disruptions [92]. Many zebrafish models of obesity have been developed in larvae and adults using overfeeding (diet-induced obesity—DIO) and/or high-fat diets (HFDs) [38,39,41,90,93,94,95]. Similarly, some genetic models have also been established [90,96,97,98].

**Overfeeding models:** Numerous obesity-inducing diet protocols have been conducted, with the difference among these models being either the nature of the food provided and/or the duration of the feeding. In 2010, Oka and colleagues overfed fish with artemia (a small shrimp used as food source in aquaculture) [99]. In their overfeeding protocol, they provided 60 mg of artemia/fish/day for 8 weeks versus 5 mg for the control [99]. At the end of their experimental procedure, the DIO fish had increased body weight, BMI and triglyceride levels and had developed hepatic steatosis.

Similar studies by Husmura and Hiramitsu showed increased visceral and subcutaneous adipose tissue volume in overfed fish and hepatic mitochondrial dysfunction [100,101]. In addition, several models of overfeeding have noted increased lipid deposition in the liver of overfed fish by Oil Red O staining, which allows the labeling of neutral lipids and cholesterol esters [38,39]. Overfed fish have also increased phosphorylation of hepatic Akt protein, a pathway involved in the development of insulin resistance [39]. The investigations of other groups have also documented that overfeeding induces higher body weight and BMI linked to the expansion of visceral and subcutaneous adipose tissue [41,95].

A different overfeeding protocol, applied for 8 months (DIO fish being fed twice as much as controls), resulted in similar disturbances: increased weight gain and steatosis of the liver (cell vacuolization), as well as an altered inflammatory response of the liver [102]. Indeed, DIO fish were unable to modulate the expression of genes involved in the inflammatory/immune system response after LPS stimulation (i.e., Toll-like receptor signaling pathway, ubiquitin-mediated proteolysis, MAPK and Jak-STAT signaling pathway, cell cycle and apoptotic genes) [102].

More recently, we established rapid and reliable models of overweight/obesity by feeding them in an “ad-libitum”-like way with conventional dry food or with a mix of artemia/conventional food for a period of 4 weeks. These models also induced higher body weight, BMI, hyperglycemia and heterogeneous liver steatosis [21,38]. Similarly, zebrafish overfed with 120 mg of commercial dry food versus 20 mg for controls for a period of 8 weeks exhibited higher body weight and BMI, hyperglycemia, glucose intolerance and increased insulin production [47].

Other experimentations have been proposed using an HFD protocol. These diets contain high amounts of fat and can be achieved using, for example, heavy whipping cream, chicken egg yolk, corn oil and lard, and ancient vegetables added (or not) to the conventional diet [39,93,103,104]. These HFD fish suffer from increased body weight including increased fat mass and hypertrophy, cardiovascular disorders, hepatic steatosis and hyperglycemia.

These different metabolic disturbances described in HFD/DIO zebrafish are also found in DIO and HFD mouse models [105,106,107,108]. Overall, these overfeeding protocols performed in zebrafish share features with human pathology: increased body weight and BMI, expansion of adipose tissue, hypertriglyceridemia, hepatic steatosis and altered expression of genes involved in lipid metabolism and inflammatory response. Hyperinsulinemia and hyperglycemia could also be observed, as well as altered expression levels of adipokines (i.e., adiponectin and leptin) and advanced glycation end products [109]. 

In conclusion, numerous models of diabetes and overweight/obesity have been established in zebrafish and have demonstrated that many mammalian (and human) features of these pathologies are shared with zebrafish (Figure 1). Interesting reviews document the different protocols of transgenic models to establish these different states of diabetes and/or obesity in zebrafish larvae and adults [90]. In this review, Salehpour and colleagues established a scoring system for type 2 diabetes in zebrafish compared to humans. Among the non-genetic models, glucose immersion as performed by [42,47] and the hyperglycemia obesity model have the highest score [90].

## 3. The Effects of Metabolic Disorders on Brain Plasticity and Function: Focus on Zebrafish and Comparative Aspects

As previously mentioned, the brain of adult zebrafish exhibits numerous neurogenic niches due to the persistence of many neural stem cells during adulthood. Zebrafish has also a strong capacity for nervous tissue regeneration [50,52,110,111]. The brain of zebrafish is also protected by a blood-brain barrier (BBB) that helps in maintaining brain homeostasis as in mammals [112]. Taken together, these intrinsic characteristics highlight the use of zebrafish to explore the deleterious effects of metabolic disorders on the central nervous system.

### 3.1. Hyperglycemia and Brain Homeostasis in Adult Zebrafish

Only a few studies have examined the impact of hyperglycemia on brain homeostasis and plasticity in adult zebrafish. While acute hyperglycemia induced by intraperitoneal injection of D-glucose (2.5 g/kg) resolves after 24 h, it nonetheless results in the upregulation of pro-inflammatory cytokines, including *il1b*, *il6*, il8 and *tnfα* [40]. In contrast, acute hyperglycemia has no effects on the expression of genes involved in BBB establishment (i.e., *claudin5a*, *zonula occludens 1a* and *1b*). Furthermore, it does not impact brain cell proliferation in the main neurogenic niches studied (OB/TEL, ventral and dorsal telencephalic domains, pretectum and hypothalamus) [40]. Further studies are needed to understand the impact of acute hyperglycemia on neuro-inflammation, with a focus on microglia reactivity and BBB leakage by performing extravasation assays.

Chronic hyperglycemia (111 mM D-glucose for 14 days) results in more severe detrimental effects on the brain. Although it does not alter the cerebral expression of pro-inflammatory genes, probably due to compensatory mechanisms, it leads to the significant upregulation of those related to BBB integrity [40]. These results obtained in adults are to be linked with studies performed in zebrafish larvae, for which chronic glucose exposure leads to defects in tectal blood-vessel patterning and neurovascular coupling, altering both vascular NO production and the number of cells in the vascular wall. It also induces a change in the neuronal calcium concentration and leads to the upregulation of GFAP, a marker of reactive gliosis expressed in NSCs in fish [113,114].

The cerebral redox balance is also altered in diabetic fish, as evidenced by increased levels of lipid peroxidation (TBARS analysis) and carbonylated brain proteins [115]. Similarly, the activity of the antioxidant enzyme superoxide dismutase (SOD) is reduced, as well as the expression of some redox-sensitive genes (*sod1* and *-2*, *gpx3a*, *nrf2*) [115]. These data were also partially corroborated in the retina of hyperglycemic fish [116] and in another hyperglycemic zebrafish model displaying modification in brain catalase activity [117].

Interestingly, chronic hyperglycemia alters neurogenesis within the major neurogenic areas of adult zebrafish (OB/TEL; ventral and dorsal telencephalic domains, pretectum and hypothalamus) [40]. It furthermore impairs the injury-induced neurogenesis process observed after a telencephalic injury [40]. This blunted regenerative capacity was also reported after caudal fin amputation in hyperglycemic fish, as previously mentioned [67].

From a behavioral perspective, hyperglycemic fish exhibit anxiety-like behavior and memory impairment, as shown by the inhibitory avoidance test [43,118]. These cognitive defects could originate from an alteration in the purinergic system [43]. Indeed, a significant decrease in brain ATP, ADP and AMP hydrolysis levels is observed in hyperglycemic fish, linked to the down-regulation of ectonucleoside triphosphate diphosphohydrolases (*entpd2a.1, -2a.2, -3* and *entpd8*) and adenosine receptors (*adora1*, *adora2aa*, *adora2ab* and *adora2b*). Interestingly, acetylcholinesterase gene expression and activity are also altered [43].

Overall, these data demonstrate that hyperglycemia in zebrafish promotes BBB alterations, neuroinflammation and oxidative stress in the central nervous system. It also leads to reactive gliosis and to impaired neurogenesis, as well as the development of cognitive defects and depressive-like behavior. All these disrupted processes reported in zebrafish are also observed in mammals during diabetes. For example, *claudin-5* and *occludin* expressions are downregulated, reflecting the increased BBB permeability observed in the brain of hyperglycemic mice [119]. Similarly, the number of microglia and reactive astrocytes in the hippocampus of hyperglycemic animals is increased, demonstrating a neuro-inflammatory state [120]. Other studies have shown the upregulation of pro-inflammatory genes (i.e., *tnfα*) associated with microglia activation in type 1 and type 2 diabetic mice [119]. The levels of antioxidant defenses and enzymes (i.e., glutathione -GSH- and glutathione peroxidase -GPX-) are decreased in the brain of diabetic rodents [121]. To date, numerous studies have found reduced hippocampal neurogenesis in diabetic rodents associated with cognitive defects and depressive behaviors [122,123,124,125,126,127,128,129].

Therefore, in both zebrafish and mammals, diabetes impacts the BBB, inflammatory and redox status, brain plasticity and cognitive functions (Figure 2). However, data should be reinforced in zebrafish to better understand the global impact of hyperglycemia on brain homeostasis, looking at, for instance, microglia reactivity and BBB physiology, as well as cell death under constitutive and brain injury conditions in hyperglycemic zebrafish.

### 3.2. Obesity and Brain Homeostasis in Zebrafish

Similar to diabetes, overweight and obesity are associated with a range of physiological disorders affecting the central nervous system. Models of overfeeding developed in zebrafish share many pathophysiological disturbances with their human counterparts, as described previously.

In zebrafish, 4-week overfeeding with a mixture of artemia and dry food induces BBB disruption, as shown by Evans blue leakage [38]. In comparison, overfeeding with only dry food results in lower BBB leakage [130], suggesting that “diet quality” may impair differentially BBB dysfunction. Similarly, an HFD protocol provided for 11 weeks with a mixture of standard food and lard (80% + 20%, respectively) results in the downregulation of genes involved in blood–brain barrier functions [103]. BBB disruption has also been associated with an increase in the number of activated microglia (amoeboid) in the ventral telencephalon and hypothalamus and a general increase in the brain expression of pro-inflammatory cytokines (*il1b*, *il6* and *tnfa*) and of the inflammatory transcription factor *nfkb* [38]. These data were corroborated in another overfed zebrafish model showing increased amoeboid and dystrophic microglia in the hypothalamus [131]. In a hybrid obesity model (high glucose/high cholesterol experimental protocol), the treated fish upregulated the cerebral expression of pro-inflammatory cytokine and apoptotic genes [132].

Interestingly, obese fish also have a disturbed brain redox balance. They exhibit higher levels of 4-HNE (4-Hydroxynonenal), a lipid peroxidation product known as a marker of oxidative stress [38]. The activities of antioxidant enzymes peroxidase and catalase in the brain are increased, suggesting the activation of the antioxidant response [38]. Meguro and colleagues also showed that HFD zebrafish exhibit disrupted expression of genes involved in antioxidant stress [103].

In general, obese fish (HFD and/or DIO) exhibit a decreased brain plasticity, as revealed by the mis-regulation of *bdnf* (brain derived neurotrophic factor) and *psd95* (postsynaptic density protein 95) genes [95,103]. The *ptn* growth factor is also significantly reduced in obese fish, while *caspase 9* gene expression is upregulated, suggesting increased cell death. Interestingly, several genes involved in β-amyloid metabolism are also modulated, suggesting links between diet and neurodegeneration [103]. This decrease in the expression of genes involved in brain plasticity parallels the consistent reduction in cell proliferation along the neurogenic niches of obese fish; this was found through PCNA immunohistochemistry and qPCR analysis (decreased expression of *pcna* and the progenitor marker *sox2*) [38,130,133,134]. Very interestingly, Stankiewicz and colleagues showed that obese fish display a decrease in the daily amplitude of central clock gene expression associated with the misalignment or decreased amplitude of daily patterns of key cell-cycle regulators (e.g., *cyclins A* and *B*, and *p20*) [134]. *Clock* genes are known to be involved in the regulation of stem cell activity and are expressed in the neurogenic niches in zebrafish [135,136], raising the question of the links between circadian clock perturbations and the decreased neurogenesis observed in obese fish. Considering neurological functions, the active avoidance test and locomotion are impaired in obese fish compared with controls [38,103,130].

Taken together, these central disruptions are also described in obese mammals. For example, HFD rodents exhibit BBB leakage associated with decreased expression of tight junctions (*claudin-5* and *occludins*) [137,138]. Similarly, HFD induces hippocampal and/or hypothalamic neuroinflammation with microglia activation and increased oxidative stress and leads to decreases in synaptic density and expression of genes involved in synaptogenesis [139,140,141,142]. Numerous studies have also highlighted the effect of a HFD on neurogenesis. For example, obese mice show decreased cell proliferation in neurogenic niches, namely, in the hypothalamus and hippocampus, associated with decreased memory and mood-related disruptive behavior [143]. Such brain alterations are strongly associated with cognitive impairments and increased anxiety [139,140,141,144,145].

Overall, these studies have demonstrated similar effects of obesity on the mammalian and fish brain, with impairment of the BBB leading to increased oxidative stress and neuro-inflammation, decreased brain plasticity including neurogenesis and altered cognitive behaviors (Figure 2). This also raises the question of the mechanisms sustaining such deleterious effects. Indeed, most zebrafish and mammalian models of obesity are hyperglycemic, suggesting that this condition could be already sufficient to impair BBB function and brain homeostasis.

### 3.3. Effects of Hypercholesterolemia on the CNS

Hypercholesterolemia is defined by high levels of cholesterol in the blood circulation [146]. Under hypercholersterolemic conditions, total cholesterol levels exceed 240 mg/dL, with LDL (low-density lipoproteins) higher than 160–190 mg/dL and HDL (high-density lipoproteins) lower than 40 mg/dL [147]. Although some gene mutations are responsible for familial hypercholesterolemia, dietary cholesterol consumption is associated with higher blood cholesterol levels [148,149]. Atherosclerosis is a subsequent result of the long-lasting elevated levels of cholesterol in the blood [150]. The blockage of a coronary artery may result in a heart attack. Similarly, at the level of the brain, vessel obstruction results in stroke.

In rodents, hypercholesterolemia has been shown to impact BBB functions [151,152]. It also results in increased neuroinflammation [153], increased cerebral oxidative stress biomarkers and decreased antioxidant activities [152]. Dietary cholesterol and hypercholesterolemia impact brain plasticity and neurogenesis [154,155]. In addition, early exposure to elevated cholesterol may be a risk factor for mild cognitive impairment, and hypercholesterolemia is an early risk factor for the development of Alzheimer’s disease [156]. Indeed, hypercholesterolemia has been shown to accelerate Aβ accumulation and tau pathology, which subsequently leads to cognitive impairment [157]. The link among hypercholesterolemia, cognitive dysfunction and Alzheimer’s disease is potentially mediated by increased neuroinflammation and oxidative stress [158].

Among the treatments that can be prescribed to decrease cholesterol are the statins. This drug family competitively inhibits the enzyme HMG-CoA reductase in the hepatic pathway that plays a central role in the production of cholesterol [159]. Several studies have documented the beneficial role of such cholesterol-lowering drugs in decreasing the risk of developing dementia and cognitive decline [158]. On the contrary, another body of evidence of high-cholesterol diets and controlled randomized trials shows no effects of statin treatment in terms of improving cognitive performance [160]. More strikingly, the food and drug administration indicated a potential effect of statins of inducing reversible cognitive impairments [158].

Therefore, despite the abundance of the available literature, the effects of statins on cognitive functions remain controversial [161,162,163]. While epidemiological evidence suggests a role for statins under neurodegenerative conditions, including vascular dementia, Alzheimer’s disease (AD) and Parkinson’s disease (PD), several large studies, as well as a number of case reports, contradict these findings [158]. One possible explanation for the contribution of statins to cognitive impairment is the inhibition of the protein geranylgeranyltransferase-1 (GGT) by statins. GGT is important for synapse formation and remodeling. Impaired synaptic plasticity in the hippocampus and reduced dendritic spine density in cortical neurons were observed in GGT-haplodeficient mice [164]. In addition, cholesterol depletion induced by prolonged statin exposure enhances neuroserpin protein aggregation [165]. Neuroserpin protein aggregates have been shown to be more numerous in patients with Alzheimer’s disease, and there is an association between neuroserpin and Aβ plaques in the brain of AD patients [166]. In addition, mitochondria are key organelles involved in the development of neurodegenerative diseases. Some data document the effects of statins on mitochondria acting on oxidative phosphorylation, generation of oxidative stress, uncoupling protein 3 concentration and interference in amyloid-β metabolism [167]. Interestingly, in vitro experiments have shown that rosuvastatin restores neurite outgrowth in hypoxic neurons by preserving mitochondrial functions and improving mitochondrial biogenesis. Interestingly, atorvastatin but not pravastatin impairs mitochondrial function in human pancreatic islets and rat β-cells [168], suggesting different effects of lipid-lowering drugs on mitochondria. Thus, it is becoming important to further investigate the contribution of the different statins to brain health.

Zebrafish is also emerging as a model for studying cholesterol metabolism and the effect of hypercholesterolemia and cholesterol-lowering drugs on the brain. Indeed, zebrafish have VLDL (very-low-density lipoprotein), LDL and HDL and their metabolism is quite similar to that of humans [169,170]. It consequently allows the study of new mechanisms and molecular pathways resulting from disorders associated with dyslipidemia [171]. There are also similarities between zebrafish and humans in intestinal cholesterol absorption [172]. Moreover, cholesterol-lowering drugs have been successfully used in zebrafish model and have shown a decrease in cholesterol levels within the tested fish. For example, the administration of ezetimibe and simvastatin reduces intestinal cholesterol levels in zebrafish [173,174]. Therefore, zebrafish could be a powerful model to uncover the implications of hypercholesterolemia and cholesterol-lowering drugs in brain homeostasis, health and cognition. However, to date, there are virtually no studies on the impact of hypercholesterolemia on brain functions in zebrafish.

## 4. Brain Dis-Plasticity and Metabolic Disorders: Molecular Mechanisms to Investigate

Neurogenesis is a tightly orchestrated process regulated by a combination of extrinsic and intrinsic factors [175,176,177]. Among them, inflammation and oxidative stress (induced during diabetes and obesity) are known to modulate the proliferation of neural stem/progenitor cells and the differentiation, migration and survival of new born cells. Zebrafish share many neurogenic signaling pathways and transcriptional regulations with mammals under both healthy and regenerative conditions [21,22,49,53,178,179,180]. However, there are not so many data highlighting the mechanisms regulating key neurogenic signaling pathways in these pathologies (diabetes and obesity) in neither fish nor mammals.

### 4.1. Notch Signaling in Mammal and Zebrafish Neurogenesis

One of the important signaling pathways that directly affects neural stem/progenitor proliferation and differentiation is the Notch 1 pathway [22,181,182,183,184,185,186,187,188,189]. This conserved signaling from drosophila to human plays a critical role in neural stem cell maintenance and neurogenesis during embryonic development and adult stages. The deregulation of Notch signaling has been implicated in many neurodegenerative diseases [185]. In mice, the conditional knock-out of this gene increases the proliferation of neural stem cells, while the constitutive expression of Notch-1 increases the number of progenitor cells [181,190]. Similarly, the pharmacological inhibition of Notch signaling in zebrafish increases the proliferation of neural stem cells [180,183,186,191].

Studies support the role of Notch signaling as a possible candidate for decreasing neurogenesis in case of metabolic disorders [192,193]. The offspring from HFD mice suffer from decreased neuronal progenitor proliferation, differentiation and synaptic plasticity, correlated with increased expression levels of Notch-1 signaling and its effector genes, namely, *Hes5* [192,193]. Furthermore, in another study, HFD mice displayed increased Notch signaling and exhibited defects in hypothalamic neurogenesis (differentiation) [194]. In these mice, the inhibition of Notch signaling or of the inflammatory transcription factor NF-kB improved hypothalamic NSC differentiation. Therefore, in HFD mice, the resulting inflammation could activate Notch signaling in neural stem cells and lead to neurogenic defects [194]

To our knowledge, in diabetic and/or obese fish, there are no clear data showing the disruption of Notch signaling. The DIO and hyperglycemic zebrafish models have increased brain inflammation associated with defective neurogenesis [38,40,130,134]. Similar to mammals, Notch signaling is an important regulator of adult zebrafish neurogenesis [182,183]. The different *notch receptors* are expressed in several neurogenic niches [183,187]. However, the links between Notch signaling and impaired neurogenesis in obese and/or hyperglycemic zebrafish has not yet been investigated. The preliminary results of our study on the expression of the target gene of Notch signaling in zebrafish, *her4.1*, did not show striking gene expression differences in the main neurogenic niches of control and obese fish (Figure 3). Such investigations can widely contribute to the understanding of Notch involvement in neurogenic disruption in the case of metabolic diseases.

In addition, it has been shown that NSCs from the embryo of pregnant diabetic mice exhibit altered expression of genes implied in the proliferation and cell fate specification, such as *delta-like 1* (a Notch ligand), *Hes1* and *Hes5* (key factors for Notch/Delta signaling) [195]. Furthermore, methylglyoxal, a highly reactive glycolytic intermediate metabolite, has been shown to regulate Notch signaling and, subsequently, neural progenitor fate [196].

Therefore, the disruption of Notch signaling in NSCs during obesity and/or diabetes may occur in zebrafish and modulate NSC proliferation and cell fate. However, these hypotheses definitively require further investigations.

### 4.2. BMP/Id1 Signaling and Cross-Talk with Notch Signaling in NSCs

Bone morphogenetic proteins (BMPs) are members of the transforming growth factor b (TGF-β) family [197], binding to transmembrane type I and type II receptors and leading to the activation of Smad proteins (Smad1/5/8). Smad1/5/8 bind to Smad4 [22,198]. This complex translocates to the nucleus and activates many target genes, including id1, the inhibitor of differentiation/DNA binding 1 [199]. In zebrafish, there are five helix–loop–helix transcriptional regulators in the Id family and four members in mice [200,201]. In both mammals and zebrafish, Id1 has overlapping and distinct functions during development and body homeostasis, controlling many cellular events, such as cell quiescence, differentiation and migration of different cell types [200].

In mammals, Id1 is an important factor regulating NSC quiescence in the mouse SVZ (a neurogenic niche), with quiescent adult NSCs strongly expressing Id1 [202]. In zebrafish, Id1 is only expressed in radial glial cells (neural stem cells) and mainly in the quiescent ones [201,203,204,205]. Interestingly, Id1 gain- and loss-of-function studies have shown that Id1 promotes the quiescence of neural stem cells, while its down-regulation allows the entry of the neural stem cell into the cell cycle [205]. The Id1 promoter regulation appears evolutionarily conserved, and BMP signaling is important for the correct expression of Id1 in zebrafish NSCs and in the regulation of their quiescent/proliferative state [204]. Id1 could promote NSC quiescence in both mouse and fish through Id1 interaction with members of the Hes/Her protein family (Notch target genes) [22,205].

Together, these data show that BMPs and Id1 are important regulators of neurogenic processes promoting neural stem cell quiescence in both mammals and fish [206]. With Notch, BMP/Id1 are also important actors of brain plasticity that should be studied in zebrafish during metabolic disorders. Indeed, in rodents, recent data suggest a role of BMPs in the development of metabolic disorders. Indeed, reduced BMP4 signaling may promote the development of obesity, insulin resistance and associated disruptions [207]. Other data show a role of BMP signaling in the regulation of feeding behavior in different mice models [208,209]. So far, BMPs play key roles in the regulation of energy balance in the brain and in adult neural plasticity [206]. It would be interesting to better understand the modulation and role of BMP/id signaling in the neurogenic niche of diabetic and/or obese models of mice and zebrafish, namely, in the hypothalamus, a key neurogenic region controlling the genesis of anorexigenic and orexigenic neurons.

### 4.3. The Role of Stress Hormone Signaling in Neurogenesis

Glucocorticoids are steroid hormones secreted mainly by the adrenal cortex. They are involved in several processes, such as metabolism [210], immune response [211], cardiovascular functions [212] and development [213]. De novo glucocorticoid synthesis also occurs in the brain, suggesting key roles of local synthesis in neurological functions [214,215]. Under normal physiological conditions, glucocorticoids are involved in adaptive responses. However, chronic exposure to glucocorticoids has been shown to be associated with a chronic stress state and several metabolic disorders, such as obesity, insulin resistance, glucose intolerance dyslipidemia and hypertension [216]. For instance, glucocorticoid levels have been shown to be upregulated in the adrenal glands in the case of obesity [217,218,219]. Indeed, the 11β-HSD1 enzyme that transforms cortisone into cortisol, the main active form of glucocorticoid, is increased in the adipose tissue of obese humans [220,221]. Similarly, diabetic individuals are also subjected to higher levels of glucocorticoids associated with chronic stress and depression [222,223]. The glucocorticoid responsive gene, *fkbp5*, has also been shown to be upregulated in the subcutaneous adipose tissue of type 2 diabetic patients, linked to the increase in glucose levels [224]. In addition, glucocorticoid levels are associated with non-alcoholic fatty liver, a hepatic disorder found in both obesity and diabetes [225].

In the central nervous system, chronic exposure to glucocorticoids increases neuro-inflammation and hippocampal neuronal damage [226]. Several studies have shown that high glucocorticoid levels are associated with decreased brain plasticity. Indeed, in vitro studies performed on embryonic neural stem/progenitor cells have confirmed that chronic exposure to glucocorticoids suppresses the differentiation and survival of NSCs affecting several signaling pathways and decreases the expression of neuronal and synaptic markers [227]. In parallel, long-term exposure leads, in vivo, to blunted hippocampal neurogenesis [228,229], which is restored through the use of glucocorticoid receptor antagonists [230]. Together, glucocorticoids increase the loss of hippocampal neurons, reduce adult neurogenesis and compromise cognitive functions, leading to an increased risk for developing Alzheimer’s disease [229,231]. This has been further corroborated by the study of diabetic mice that display higher levels of corticosteroids and have increased tau protein phosphorylation correlated with memory impairments [232]. Overall, these data support the fact that chronic exposure to glucocorticoids negatively affects brain homeostasis and plasticity, promoting behavioral change and increasing the risk factors for cognitive defects and neurodegenerative disease.

In zebrafish, the inter-renal gland ensures the function of the adrenal gland found in mammals and secretes glucocorticoids [233]. In addition, the brain is able to perform de novo steroidogenesis and glucocorticoid synthesis [234,235,236,237]. In a zebrafish model of obesity associated with hyperglycemia, the expression of the glucocorticoid responsive gene *fkbp5* was up-regulated but did not reach a significant level [130]. Unpublished data from our laboratory using a stronger model of overfeeding also demonstrated an increase in the cerebral expression of *fkbp5*. Together, these data suggest that the peripheral and/or locally produced glucocorticoids may be increased under overfeeding conditions. Our preliminary data for measuring glucocorticoid levels in the brain of obese/hyperglycemic fish tend to show an increase in glucocorticoid levels. Further investigations are consequently required to understand whether glucocorticoids are indeed significantly increased during zebrafish obesity and diabetes or not and what are their roles in brain plasticity in these pathologies in zebrafish.

Many other factors well known to modulate and regulate neurogenesis should also be investigated, such as Wnt signaling, SHH, endothelial factors, inflammation, oxidative stress and steroids other than glucocorticoids. Figure 4 provides a schematic overview of some pathways to be investigated in the brain and NSCs under metabolic disease conditions.

## 5. Conclusions

Under diabetic and obese conditions, the central nervous system is greatly affected through the disruption of the BBB and subsequent neuro-inflammation and oxidative stress. In both mammalian and zebrafish models, metabolic disorders induce decreased brain plasticity, impaired cognitive functions and abnormal behaviors. However, the precise cellular and molecular mechanisms sustaining such impairments are not well understood. In this review, we aim to demonstrate that zebrafish is an alternative model to study the impact of metabolic disorders on brain homeostasis and NSC activity. Several groups have developed successful metabolic models of zebrafish, mimicking overweight, obese and hyperglycemic/diabetic states. Indeed, zebrafish under appropriate experimental procedures can show many features of human metabolic disruption: higher body weight and BMI, expansion of visceral and subcutaneous adipose tissues, liver steatosis, disturbed lipidic profiles (LDL, HDL, body cholesterol and triglycerides), insulin resistance and hyperglycemia. Furthermore, zebrafish is well-recognized as an interesting model for studying brain plasticity, including homeostatic and regenerative neurogenesis [21,22,49,55,180,187,205,238,239]. Consequently, this model is appropriate to further investigate the role of metabolic disorders in adult neurogenesis. Now, further explorations of the molecular mechanisms and signaling pathways disrupted in NSCs should be performed under metabolic disorder conditions.

## Figures and Tables

**Figure 1 ijms-23-05372-f001:**
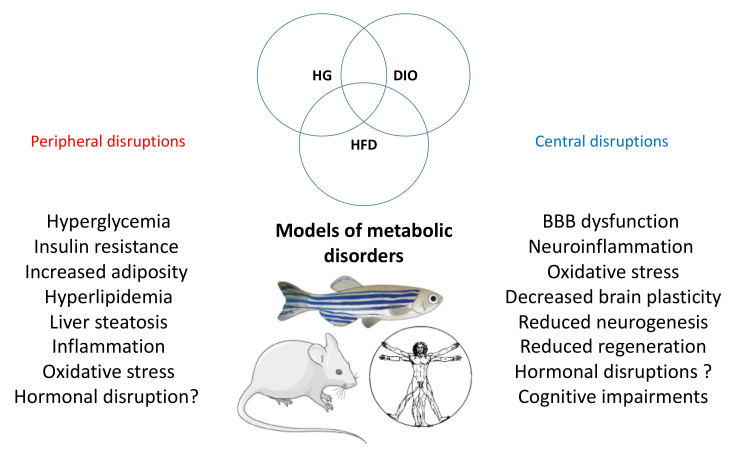
General overview of the peripheral and central disruptions induced by metabolic disorder. The main peripheral and central disruptions are observed in the different models of metabolic disorders (hyperglycemia (HG), DIO and HFD) in fish and rodents. These pathological processes are also found in humans.

**Figure 2 ijms-23-05372-f002:**
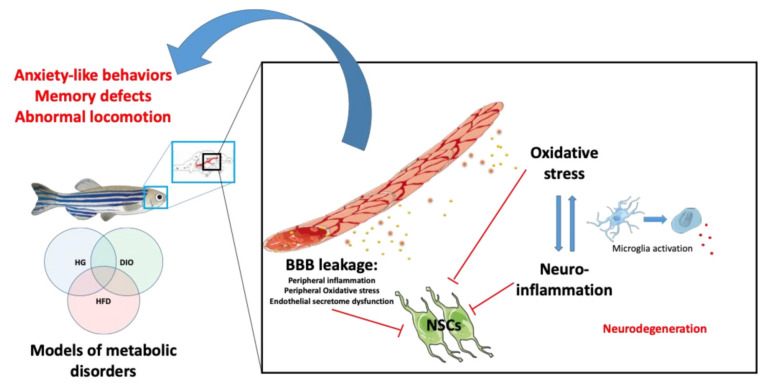
Peripheral and central mechanisms impacting brain homeostasis in metabolic diseases. Under hyperglycemic (HG), DIO and HFD conditions, BBB breakdown occurs and leads to central oxidative stress and neuroinflammation through the activation of microglia (switch from ramified to ameboid state). It can result in neurodegeneration. In addition, metabolic disorders could lead to impaired secretion of endothelial factors that could, in synergy with the disrupted peripheral and central factors, impair neural stem cell (NSC) activity. The ultimate consequence of such disruptions is the development of cognitive impairments (locomotion, anxiety, memory) and neurodegenerative diseases.

**Figure 3 ijms-23-05372-f003:**
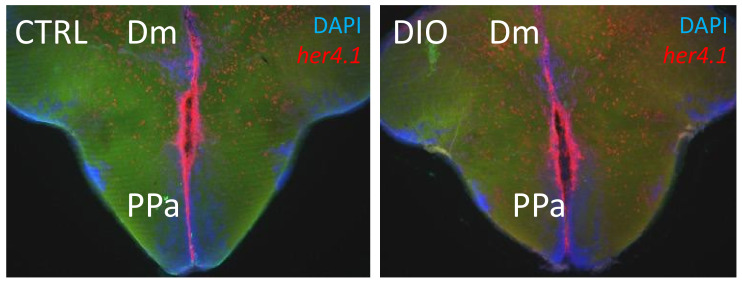
*her4.1* in situ hybridization on brain section of CTRL and DIO fish. Preliminary data on DIO fish (4 weeks) did not show striking differences in the expression of the Notch target gene, *her4.1* (red), in CTRL and DIO fish. Cell nuclei were counterstained with DAPI (blue). Dm: dorsomedian telencephalon. PPa: anterior part of the preoptic area.

**Figure 4 ijms-23-05372-f004:**
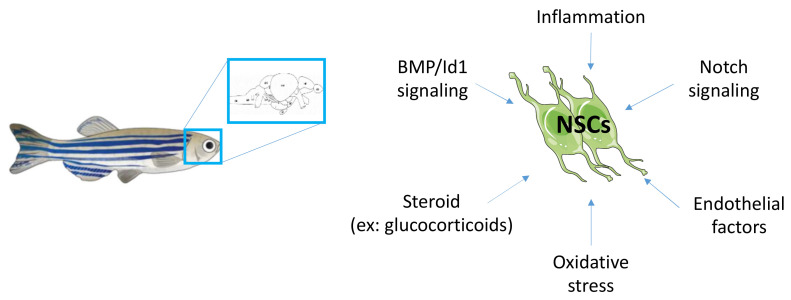
Mechanisms to be investigated regarding the misregulated neurogenic signaling pathways under conditions of metabolic diseases.

## Data Availability

The data presented in this study are available on request from the corresponding author.

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
