# Peer review of "Zebrafish: A New Promise to Study the Impact of Metabolic Disorders on the Brain"

_ijms, 2022, doi:10.3390/ijms23105372_

Round 1

Reviewer 1 Report

The Authors gather in this review the scientific evidences establishing that metabolic disorders can alter brain homeostasis, with a particular focus to obesity and diabetes. The work is very well organized, but in my opinion, it could be completed with a collection of data on the effects of hypercholesterolemia on CNS, and the potential paradoxical role of the cholesterol-lowering drugs (statins) in the induction of mitochondrial dysfunction underlying the onset of sporadic forms of protein misfolding in neurodegenerative diseases,  including the sporadic form of CJD, Alzheimer's disease and Amyloidosis.

Author Response

Dear Reviewer,

Thanks for your positive comments and suggestions. We add a 3.3 subpart entitled Effects of hypercholesterolemia on the CNS. In this part, we briefly discuss the impact of hyperglycemia on brain homeostasis and plasticity using mainly mammalian literature given that almost no data are available in zebrafish. We also mention the role of statins on some brain function and mitochondrial dysfunction.

We hope that these changes will satisfy the requirements

Best

Reviewer 2 Report

This is a nice review of the interplay between metabolism and neurogenesis (and pathology) in fish and how it relates to similar aspects of human biology. It is well written, of general interest, and I do not have any major issues with the review. 

Author Response

We thanks Reviewer for these comments

Best